# A Septin Cytoskeleton-Targeting Small Molecule, Forchlorfenuron, Inhibits Epithelial Migration via Septin-Independent Perturbation of Cellular Signaling

**DOI:** 10.3390/cells9010084

**Published:** 2019-12-29

**Authors:** Lei Sun, Xuelei Cao, Susana Lechuga, Alex Feygin, Nayden G. Naydenov, Andrei I. Ivanov

**Affiliations:** 1Department of Inflammation and Immunity, Lerner Research Institute of Cleveland Clinic Foundation, Cleveland, OH 44195, USA; sunl3@ccf.org (L.S.); caox@ccf.org (X.C.); lechugs@ccf.org (S.L.); naydenn@ccf.org (N.G.N.); 2School of Nursing, Virginia Commonwealth University School of Nursing, Richmond, VA 23298, USA; afeygin@vcu.edu

**Keywords:** septin cytoskeleton, epithelial cells, migration, wound healing, hepatocyte growth factor, epithelial barrier, off-target effects

## Abstract

Septins are GTP-binding proteins that self-assemble into high-order cytoskeletal structures, filaments, and rings. The septin cytoskeleton has a number of cellular functions, including regulation of cytokinesis, cell migration, vesicle trafficking, and receptor signaling. A plant cytokinin, forchlorfenuron (FCF), interacts with septin subunits, resulting in the altered organization of the septin cytoskeleton. Although FCF has been extensively used to examine the roles of septins in various cellular processes, its specificity, and possible off-target effects in vertebrate systems, has not been investigated. In the present study, we demonstrate that FCF inhibits spontaneous, as well as hepatocyte growth factor-induced, migration of HT-29 and DU145 human epithelial cells. Additionally, FCF increases paracellular permeability of HT-29 cell monolayers. These inhibitory effects of FCF persist in epithelial cells where the septin cytoskeleton has been disassembled by either CRISPR/Cas9-mediated knockout or siRNA-mediated knockdown of septin 7, insinuating off-target effects of FCF. Biochemical analysis reveals that FCF-dependent inhibition of the motility of control and septin-depleted cells is accompanied by decreased expression of the c-Jun transcription factor and inhibited ERK activity. The described off-target effects of FCF strongly suggests that caution is warranted while using this compound to examine the biological functions of septins in cellular systems and model organisms.

## 1. Introduction

The cytoskeleton is a major determinant of the architecture and function of eukaryotic cells. It is composed of various filaments formed via self-assembly and the polymerization of specialized structural proteins [1,2,3]. There are four types of cytoskeletal structures in eukaryotic cells: actin filaments, microtubules, intermediate filaments, and septin complexes. These cytoskeletal elements play crucial roles in regulating homeostatic and specialized functions of different cells. Such functions include regulation of cell shape and size, cell division, migration, vesicle trafficking, cell-cell interactions, receptor signaling, etc. [1,2,3]. Abnormal architecture and dynamics of different cytoskeletal structures are linked to the development of various diseases, most notably cancer and inflammation [4,5,6,7].

Crucial advances in understanding the organization and cellular functions of the cytoskeleton were driven by the discovery of cell permeable small molecules that selectively bind to, and alter, the dynamics of different cytoskeletal elements [8]. For example, deciphering the roles of actin filaments and the non-muscle myosin II (NM II) motor in various cellular processes was greatly accelerated through the use of actin filament-disrupting drugs, cytochalasins [9] and latrunculins A/B [10,11], a filament stabilizer, jasplakinolide [12], and NM II inhibitor, blebbistatin [13]. Examples of important pharmacological tools selectively targeting microtubules include nocodazole and taxol, which inhibit microtubule assembly and disassembly, respectively [14,15]. The aforementioned cytoskeletal drugs appear to be highly specific to their targets since no major off-target effects are reported after extensive investigation of these compounds. However, in addition to these success stories, there are many examples of different cytoskeleton-modulating compounds that turned out to not be as specific as suggested during their initial discovery and characterization. For example, wiskostatin, initially developed as a blocker of N-WASP-dependent actin polymerization [16], was subsequently shown to also be a potent ATP-depleting agent [17]. Another example of a non-specific compound is 2,3 butanedione monoxime, used to inhibit NM II ATPase activity [18,19], but appeared to disrupt the actin filament assembly via NM II-independent mechanisms [20,21]. Furthermore, a small molecule, miuraenamide A, was originally characterized as a novel actin filament-stabilizing drug [22]. However, a subsequent study revealed its profound effects on protein expression that includes decreasing the levels of components of the Wnt signaling pathway [23]. These examples illustrate that caution should be used in interpreting the cellular effects of new or poorly characterized cytoskeletal drugs; proposed mechanisms of their actions should be verified using more specific genetic tools.

Septin cytoskeleton is the least studied cytoskeletal type in eukaryotic cells. In mammals, septin filaments and rings are assembled by hetero-oligomerization of thirteen different members of a family of GTP-binding proteins [1,24,25,26]. Septin filaments interact with membrane phospholipids and other cytoskeletal elements, thereby acting as important cellular scaffolds controlling the shape and positioning of different intracellular organelles [1,24,25,26]. The structure and functions of the septin cytoskeleton have been studied using different experimental approaches. One such approach utilizes forchlorfenuron (FCF), a synthetic small molecule suggested to selectively target septin filaments, affecting their architecture and dynamics [27,28]. In silico docking proposes that FCF stabilizes septin filaments by interacting with the nucleoid-binding pocket of septin monomers, thereby preventing GTP binding and hydrolysis [29]. A number of studies have used FCF treatment to interrogate the functional roles of septin filaments in different mammalian cells. FCF was shown to interfere with cell proliferation [27,30], block cell migration and invasion [27,30,31], attenuate formation of the epithelial barrier [32], modulate growth factor receptor trafficking and signaling [33], block store-operated calcium entry [34,35], and inhibit synaptic transmission [36,37,38]. Targeting the septin cytoskeleton by FCF was frequently complemented by genetic downregulation of septin expression [31,32,34,35,38,39]. However, in several studies, FCF was the only tool used to perturb septin filament organization and dynamics [30,33,36,37].

FCF appears to be active at relatively high concentrations, 50–500 µM, increasing the probability of off-target effects. A study in budding yeast cells demonstrated that FCF was able to inhibit the proliferation of septin deficient mutants, thus highlighting septin-independent cellular activities of this compound [40]. The possibility of off-target effects of FCF in other experimental models was suggested by the same group [40,41] but has never been proven experimentally. Since FCF is frequently used to probe septin functions in different mammalian cells, it is important to understand its specificity in such experimental systems. In the present study, we found that FCF inhibits the spontaneous, as well as hepatocyte growth factor (HGF)-induced, motility of human colonic and prostate epithelial cells, and disrupts the integrity of the epithelial barrier in colonic epithelial cell monolayers. Surprisingly, these inhibitory effects of FCF were evident even after the genetic disruption of the septin cytoskeleton, suggesting septin-independent functions of FCF in human epithelial cells. One such septin-independent FCF activity involves the modulation of intracellular signaling via the suppression of c-Jun expression and ERK signaling.

## 2. Materials and Methods

### 2.1. Antibodies and Other Reagents

The following primary polyclonal (pAb) and monoclonal (mAb) antibodies were used to detect cytoskeletal and signaling proteins: anti-SEPT2, SEPT9 pAbs, and anti-α-tubulin mouse mAb (Millipore-Sigma, St. Louis, MO, USA); anti-SEPT2, SEPT6, and SEPT8 pAbs (Proteintech, Rosemont, IL, USA); anti-SEPT7 pAb (Santa Cruz Biotechnology, Dallas, TX, USA); anti-SEPT11 pAb (Abcepta, San Diego, CA, USA); anti-Erk1/2 and phospho-Erk1/2, c-Jun, phospho-c-Jun, Akt1, phospho-Akt, phospho-SAPK/JNK and Src rabbit mAbs, anti-phospho-Src, FAK, phospho-FAK, SAPK/JNK, and GAPDH pAbs (Cell Signaling, Danvers, MA, USA); anti-FAK and Paxillin mouse mAbs (BD Biosciences, San Jose, CA, USA); anti-phospho Paxillin pAb (Thermo-Fisher Scientific, Waltham, MA, USA); Alexa Fluor-488-conjugated phalloidin and Alexa Fluor-555-labeled donkey-anti-mouse secondary antibody were obtained from Thermo-Fisher. Horseradish peroxidase (HRP)-conjugated goat-anti-rabbit and anti-mouse secondary antibodies were acquired from Bio-Rad Laboratories (Hercules, CA, USA). Human recombinant HGF was purchased from R&D Systems, (Minneapolis, MN, USA); *N*-(2-Chloro-4-pyridyl)-*N*′-phenylurea (forchlorfenuron, FCF) was purchased from Millipore-Sigma (St. Louis, MO, USA). All other reagents were of the highest grade and obtained from either Thermo-Fisher or Millipore-Sigma.

### 2.2. Cell Culture

HT-29 cf8, a well-differentiated clone of HT-29 human colonic epithelial cells [42,43] was provided by Dr. Judith M. Ball (College of Veterinary and Biomedical Sciences, Texas A&M University, College Station, TX, USA). DU145 human prostate epithelial cells were obtained from American Type Culture Collection (Manassas, VA, USA), and 293FT cells were obtained from Thermo-Fisher. HT-29 and 293FT cells were grown in DMEM supplemented with 10% FBS, 15 mM HEPES, non-essential amino-acids, and penicillin-streptomycin antibiotic. DU145 cells were grown in RPMI media supplemented with 10% FBS, 5 mM pyruvate, 15 mM HEPES, and penicillin-streptomycin antibiotic. Cells were grown and propagated in T75 flasks and were plated on either 6-well plates, collagen-coated Transwell chambers (Millipore-Sigma), or collagen-coated coverslips for biochemical and functional studies.

### 2.3. Generation of Septin 7 Knockout Lines Using CRISPR-Cas9 Gene Editing

Guide RNA sequences for CRISPR/Cas9 were designed with a CRISPR design web site (http://crispr.mit.edu/), provided by the Feng Zhang Lab (Broad Institute of the Massachusetts Institute of Technology and Harvard University, Boston, MA, USA). In order to construct plasmids for CRISPR-mediated SEPT7 gene knockout, the lentiCRISPR v2 (Addgene: #52961) vector was used as a backbone. The vector was digested with a BsmBI restriction endonuclease and hybridized oligos were ligated to a single guide RNA (sgRNA). The following sgRNA sequences were used in the study: SEPT7-sgRNA1: CACCGCAGCAACAGAAGAACCTTGA; SEPT7-sgRNA2: CACCGCTGGAGAATACAAATCTGTG; control sgRNA: CACCGGACCGGAACGATCTCGCGTA. Lentiviruses were produced in 293FT cells using helper plasmids pCD/NL-BH*DDD (Addgene: #17531) and pLTR-G (Addgene: #17532) transfected with TransIT^®^-293 Transfection Reagent (Mirus Bio, Madison, WI, USA). Stable SEPT7-depleted HT-29 cells were generated by transduction with the SEPT7 sgRNAs containing lentiviruses and subsequent puromycin selection (5 μg/mL) for 7 days.

### 2.4. RNA Interference

SEPT7 expression was transiently downregulated in DU145 cells using gene-specific small interfering (si) RNAs, as previously described [44,45,46]. A siGENOME SEPT7 SmartPool (Horizon Discovery, Cambridge, UK) was used to downregulate SEPT7 expression, whereas a non-targeting duplex #1 was used as a control. Cells were seeded in 6-well plates at approximately 60% confluence and transfected with corresponding siRNAs using DharmaFect 1 transfection reagent [46,47]. The final siRNA concentration in the final transfection mixture was either 50 nM or 100 nM. Cells were utilized for experiments on days 3 and 4 post-transfection.

### 2.5. Immunoblotting Analysis

Total cell lysates were obtained by homogenizing cells with RIPA cell lysis buffer (20 mM Tris, 150 mM NaCl, 2 mM EDTA, 2 mM EGTA, 1% sodium deoxycholate, 0.1% SDS, 1% Triton X-100), supplemented with phosphatase inhibitor cocktails 2 and 3 (1:200) and protease inhibitor cocktail (1:100) (Millipore-Sigma). The homogenized samples were cleared by centrifugation, mixed with an equal volume of 2× SDS sample buffer, and boiled. Total cell lysates were separated by SDS-polyacrylamide gel electrophoresis with 10–20 µg of protein loaded into each well. The separated proteins were then transferred by standard electroblotting technique onto nitrocellulose membranes. After transfer, membranes were incubated with primary and HRP-conjugated secondary antibodies, and the proteins were visualized using standard enhanced chemiluminescence reagents and x-ray film. Protein expression was quantified by densitometry using Epson Perfection V500 photo scanner (Epson America Inc. Long Beach, CA, USA) and ImageJ software (National Institute of Health, Bethesda, MD, USA) of three immunoblot images, each representing an independent experiment. Data are presented as normalized values assuming the expression level in control sgRNA, or siRNA-treated groups as 1. Statistical analysis was performed with row densitometric data using GraphPad Prism 6.01 (San Diego, CA, USA).

### 2.6. Immunofluorescence Labeling and Confocal Microscopy

Control and SEPT7-depleted epithelial cells cultured on collagen-coated coverslips were fixed in 4% paraformaldehyde and permeabilized with 0.5% Triton-X100 at room temperature. Fixed samples were blocked for 60 min in HEPES-buffered Hanks’ balanced salt solution (HBSS) containing 1% bovine serum albumin, followed by a 60-min incubation with anti-α-tubulin antibody. Samples were then washed and incubated with Alexa-Fluor-488–conjugated phalloidin and Alexa-Fluor-555–conjugated donkey anti-mouse secondary antibodies for 60 min, rinsed with blocking buffer, and mounted on slides with ProLong Antifade mounting reagent (Thermo-Fisher). Immunolabeled cell monolayers were imaged using a Leica SP8 confocal microscope (Wentzler, Germany). The Alexa Fluor 488 and 555 signals were acquired sequentially in frame-interlace mode, to eliminate cross talk between channels. Images were processed using Adobe Photoshop. The images shown are representative of at least three experiments, with multiple images taken per slide.

### 2.7. Scratch Wound Assay

Epithelial cells were plated in 6-well plates and grown for 5 days until confluency. A pipette tip was used to make a thin scratch wound in the cell monolayer. The bottom of the well was marked to define the initial position of the wound, and the monolayers were supplied with fresh cell culture media. The images of a cell-free area at the marked region were acquired at the indicated times after wounding, using a Leica DMi8 inverted bright field microscope equipped with a camera. The width of the wound area, along the established marks, was measured using ImageJ software. For biochemical experiments, multiple wounds were created in cell monolayers using a Cell Comb™ Scratch kit (Millipore-Sigma).

### 2.8. Measurement of Epithelial Barrier Permeability

Transepithelial electrical resistance (TEER) of cultured epithelial cell monolayers was measured using an EVOMX volt-ohm meter (World Precision Instruments, Sarasota, FL, USA). The resistance of cell-free collagen-coated filters was subtracted from each experimental point. An in vitro dextran flux assay was performed, as previously described [43,47]. Briefly, control and SEPT7-deficient HT-29 cell monolayers differentiated on transwell filters were apically exposed to 1 mg/mL of fluorescein isothiocyanate-dextran (4000 Da, Millipore-Sigma) in HBSS. After 4 h incubation, HBSS samples were collected from the lower chamber, and fluorescein isothiocyanate fluorescence intensity was measured using a SpectraMax M2 Microplate Reader (Molecular Devices, San Jose, CA, USA) at excitation and emission wavelengths of 485 and 544 nm, respectively. After subtracting the fluorescence of dextran-free HBSS, relative intensity was calculated using GraphPad Prism 6.01.

### 2.9. Statistics

All data are expressed as means ± standard error (SE) from three biological replicates. Statistical analysis was performed using a one-way ANOVA to compare obtained numerical values in the control and two experimental groups (knockout with two different SEPT7 sgRNAs). If the ANOVA test showed significant differences, a post-hoc t-test was used to compare the difference between the control and each SEPT7-depleted group. *p* values < 0.05 were considered statistically significant.

## 3. Results

### 3.1. FCF Attenuated Spontaneous, and Stimulated, Migration of Human Epithelial Cells

The effects of FCF in human epithelial cells were examined using both spontaneous and HGF-stimulated cell migration as major functional readouts, since inhibition of cell motility with either FCF treatment, or genetic depletion of different septins, has been previously reported [27,30,31]. Well-differentiated HT-29 cf8 human colonic epithelial cells and DU145 human prostate epithelial cells were used in this study; their spontaneous and HGF-induced migration was investigated using a classical scratch wound healing assay. Our pilot experiments demonstrated different velocities of wound healing for these two cell lines, with HT-29 cells migrating much slower, compared to DU145 cells. Thus, the motility of HT-29 and DU145 cell monolayers was examined over different time intervals, up to 24 h and 8 h, respectively, to allow for substantial wound closure. FCF was added at a final concentration of 50 µM, which is at the lowest end of the already established effective concentration range for this compound (50–500 µM). Epithelial cell monolayers were pre-incubated for 2 h with either FCF or vehicle (DMSO), wounded, and allowed to migrate in the presence of either FCF or vehicle for the indicated times. In HT-29 cell monolayers, FCF significantly attenuated spontaneous cell migration (Figure 1). Furthermore, this compound completely blocked the increase in cell migration caused by HGF (Figure 1). By contrast, FCF treatment did not affect spontaneous wound healing in DU145 cell monolayers but significantly attenuated their HGF-induced motility (Figure 2).

### 3.2. Downregulation of Septin 7 Expression Triggered the Loss of Other Septin Proteins in Epithelial Cells

Next, we sought to investigate whether or not the observed inhibition of cell migration caused by FCF treatment is mediated by dysfunction of the septin cytoskeleton. This question was addressed by comparing the effects of FCF on control epithelial cells and cells with genetic disruption of the septin cytoskeleton. The septin cytoskeleton was disrupted via downregulation of septin 7 (SEPT7) expression, which is known to destabilize many other septin proteins and trigger their degradation [48,49]. Two different approaches were used for SEPT7 downregulation: a stable CRISPR/Cas9 dependent knockout of this protein in HT-29 cells, and transient, siRNA-mediated knockdown of SEPT7 in DU-145 cells. A side-by-side comparison of different techniques for SEPT7 depletion helps to minimize possible influences of distinct non-specific cellular responses to gene knockout and knockdown procedures. Both CRISPR/Cas9-mediated knockout and siRNA-mediated knockdown resulted in a marked decrease in SEPT7 protein levels (Figure 3). Consistent with our expectations, loss of SEPT7 resulted in dramatic expressional downregulation of other major septins (SEPTs 2, 6, 8, 9, 11) in both HT-29 and DU145 cells (Figure 3). These results indicate a global disruption of the septin cytoskeleton in SEPT7-depleted epithelial cells.

### 3.3. FCF Attenuated Migration and Impaired Barrier Properties of Epithelial Cell Monolayers Lacking the Septin Cytoskeleton

Next, we determined if the disruption of the septin cytoskeleton affects cellular responses to FCF. Surprisingly, FCF treatment significantly attenuated both spontaneous wound healing (Figure 4A–C) and HGF-stimulated migration of septin-depleted HT-29 cells (Figure 4D–F). Interestingly, the magnitude of FCF-dependent inhibition of wound healing appears to be similar to, or even higher, in septin-depleted HT-29 cells, compared to their controls. Consistently, FCF exposure inhibited HGF-induced migration of SEPT7-depleted and control DU145 cells, with similar efficiency (Figure 5). In order to expand upon the biological relevance of our findings, we sought to examine the effects of FCF on another important epithelial function. Specifically, we focused on the integrity of the paracellular barrier because it is one of the most characteristic features of differentiated epithelial cells that is known to be regulated in a septin-dependent fashion [32]. Since DU145 cell monolayers did not develop a robust barrier (data not shown), these experiments were performed only in HT-29 cells. Confluent, well-differentiated HT-29 cell monolayers were treated with FCF for different time intervals (up to 24 h), and their paracellular permeability was determined by measuring TEER and trans-monolayer flux of FITC-conjugated dextran. FCF treatment of control cell monolayers resulted in decreased TEER and increased dextran flux, thereby indicating enhanced permeability to small ions and large uncharged molecules, respectively (Figure 6A,B). SEPT7-depleted HT-29 cell monolayers displayed increased paracellular permeability even without drug treatment. However, FCF treatment further decreased TEER and markedly increased dextran flux in septin-depleted cells (Figure 6A,B). The magnitude of FCF-induced increase in dextran flux was higher in cell monolayers with disrupted septin cytoskeleton, compared to control epithelial cells (Figure 6B).

Overall, this data strongly suggests that FCF inhibits migration and disrupts barrier properties of epithelial cell monolayers via mechanisms independent of the septin cytoskeleton.

### 3.4. FCF Treatment Inhibited Intracellular Signaling Events

Finally, we sought to elucidate the mechanisms by which FCF affects the migration and permeability of epithelial cell monolayers. One possibility would involve alterations to the actin or microtubule cytoskeleton, which are known to be essential for epithelial wound healing and barrier assembly [50,51,52,53]. However, immunofluorescence labeling and confocal microscopy did not show significant changes to the organization of actin filaments and microtubules at the migrating edge of HT-29 cell monolayers treated with either FCF or vehicle (Figure 7). Similar results were obtained after fluorescence labeling of vehicle and FCF-treated SEPT7-depleted HT-29 cell monolayers (data not shown). We also performed an extensive immunoblotting analysis to examine the activation status of crucial promigratory signaling pathways. Interestingly, our data demonstrated that FCF inhibited expression and activation of the c-Jun transcription factor and attenuated activation (phosphorylation) of a mitogen-induced kinase, ERK in migrating HT-29 cells (Figure 8A–D).

The described effects of FCF on promigratory signaling pathways were detected in both control and septin deficient cell monolayers (Figure 8). Furthermore, loss of the septin cytoskeleton resulted in increased Src phosphorylation that was not mimicked or affected by FCF treatment (Figure 8A,E). Overall, this data indicates that FCF can modulate different signaling cascades in epithelial cells in a septin-independent fashion.

## 4. Discussion

The septin cytoskeleton, which is the fourth cytoskeletal element of eukaryotic cells, has a number of homeostatic and specialized functions in different tissues [1,24,25,26]. In contrast to the extensive toolbox for probing the structure and dynamics of either actin cytoskeleton or microtubules, experimental tools to probe septin functions are limited. For example, only one small molecular compound, FCF, has been developed to interrogate septin filament functions in different experimental systems. Data obtained by probing the septin cytoskeleton with FCF were used to develop several hypotheses regarding the cellular/physiological functions of septins. These hypotheses include: roles of the septin cytoskeleton in regulating cancer cell migration and proliferation [27,30,31], neurotransmitter release in motor neuron endings [36,37,38], mediating osteoclastic bone resorption [39], and controlling calcium channel activity [34,35]. Convincing evidence demonstrates the potent effects of FCF on septin filament organization in vitro and in vivo. In cell-free systems, concentrations in the micromolar range of this compound stimulated the lateral association and aggregation of purified septin filaments [27] and induced polymerization of septin complexes isolated from *Schistosoma mansoni* [54]. FCF added to *Saccharomyces cerevisiae* and filamentous fungus *Ashbya gossypil* caused the reversible formation of abnormal and mislocalized long septin fibers [28,55]. Finally, in mammalian epithelial cells, FCF increased the length and thickness of different septin structures [27,38]. This data firmly established the ability of FCF to modulate the structure and function of the septin cytoskeleton. The question remains as to whether the septin cytoskeleton is the only or even the primary target for FCF?

Interestingly, previous studies in non-mammalian systems did suggest possible off-target effects of FCF. For example, this synthetic plant cytokinin has been extensively used in agriculture to increase fruit size in higher plants not expressing septins. In plants, FCF actions have been explained as inhibiting cytokinin oxidase/dehydrogenase [56]. In budding yeasts, low concentrations of FCF significantly attenuated cell growth without altering the organization of septin filaments, whereas in fission yeasts, the inhibitory effects of FCF on cell morphology and proliferation were not phenocopied by septin mutants [40,41]. In these systems, FCF addition resulted in mitochondrial fragmentation, which could reflect the stimulation of a general stress response [40].

The present study shows that FCF could inhibit two important functions of human epithelial cell monolayers: collective cell migration, and maintenance of the epithelial barrier even under conditions of global downregulation of the septin cytoskeleton (Figure 3, Figure 4, Figure 5 and Figure 6). Our results, demonstrating good agreement with the aforementioned yeast studies, strongly suggest that the described FCF activities represent off-target effects of this compound. Interestingly, FCF treatment either mimicked or exaggerated the effects of genetic disruption of the septin cytoskeleton, or exerted functional activity that was not recapitulated by septin depletion. For example, disruption of the septin cytoskeleton by CRISPR/Cas9-mediated knockout of SEPT7 resulted in the impairment of the epithelial barrier, exaggerated by FCF treatment (Figure 6). By contrast, genetic disruption of the septin cytoskeleton did not affect either spontaneous or HGF-induced epithelial wound healing, while FCF treatment inhibited these processes (Figure 4 and Figure 5). While we did not intend to fully dissect the molecular mechanisms that underline the observed off-target effects of FCF in human epithelial cells, our data suggest that this compound could affect crucial cellular signaling pathways. One such pathway involves the c-Jun transcriptional factor. Indeed, under our experimental conditions, FCF consistently decreased the expression and phosphorylation of c-Jun protein in a septin-independent fashion (Figure 8). Since c-Jun is a component of key AP-1 transcriptional factor, controlling many vital cellular functions [57], its downregulation could explain the observed inhibitory effects of FCF on epithelial cell migration and barrier assembly. This is not the first reported case of FCF affecting protein expression. Previous studies observed that this compound blocked the expression and transcriptional activity of hypoxia-induced factor-1α in human prostate cancer cells [30] and inhibited expression of Gata7 and Sox9 transcription factors in rat cardiomyocytes [58]. Such FCF effects were not validated with genetic manipulation of septin expression, and it remains unclear if they are mediated by the septin cytoskeleton. Another possible septin-independent target of FCF involves inhibition of ERK signaling (Figure 8). ERK is known to be essential for cell migration [59], and inhibition of ERK is likely to contribute to the attenuated wound healing observed in FCF-treated epithelial cell monolayers.

In conclusion, our study demonstrates that a widely used pharmacological modulator of the septin cytoskeleton, FCF, potently inhibits migration and impairs the barrier properties of human epithelial cells in a septin cytoskeleton-independent fashion. This is the first evidence of non-septin (i.e., off-target) effects of FCF in mammalian cells. The described off-target effects neither preclude the ability of FCF to disrupt the septin cytoskeleton, nor do they suggest discontinuing the use of this compound as a pharmacological tool to probe septin function. However, our results discourage the use of FCF alone and suggest that data obtained using this compound should be interpreted with caution and, thus, should be verified using more specific genetic approaches to interfere with septin expression and functions.

## Figures and Tables

**Figure 1 cells-09-00084-f001:**
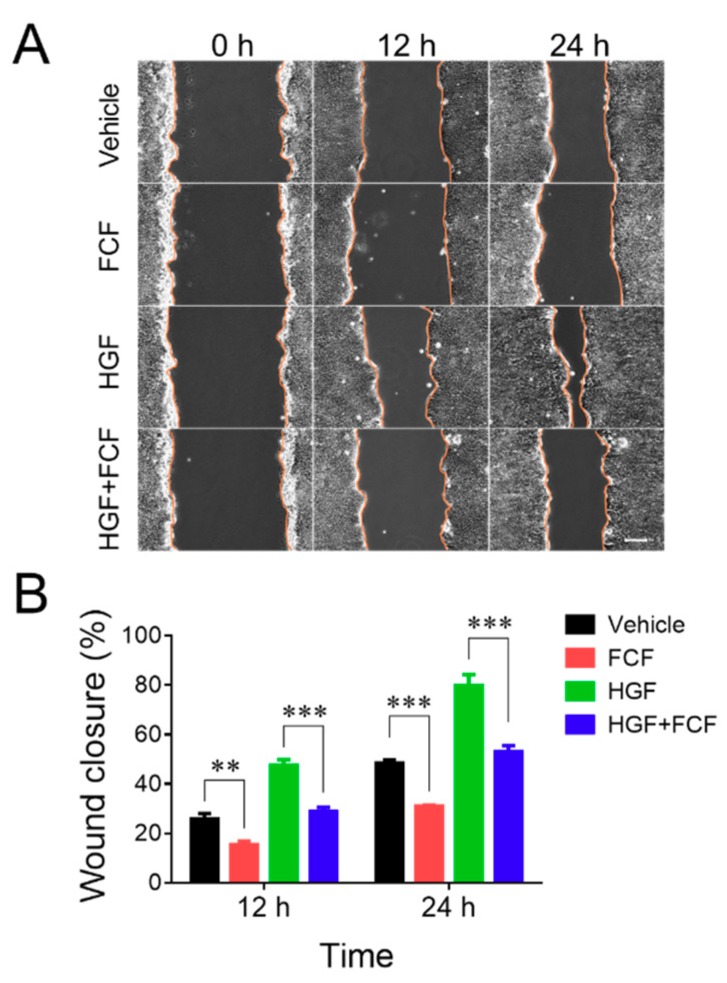
Forchlorfenuron attenuates the spontaneous and hepatocyte growth factor-induced migration of colonic epithelial cells. Confluent HT-29 cell monolayers were pretreated for 2 h with either forchlorfenuron (FCF, 50 μM), or vehicle (DMSO), and wounded. Spontaneous and hepatocyte growth factor (HGF, 25 ng/mL)-induced wound closure with, or without, FCF was examined at the indicated time points. (**A**) Representative images of wounded HT-29 cell monolayers. (**B**) Quantitation of wound closure during 12 and 24 h of cell migration. Data are presented as a mean ± SE (*n* = 5); ** *p* < 0.01, *** *p* < 0.001. Scale bar, 100 µm.

**Figure 2 cells-09-00084-f002:**
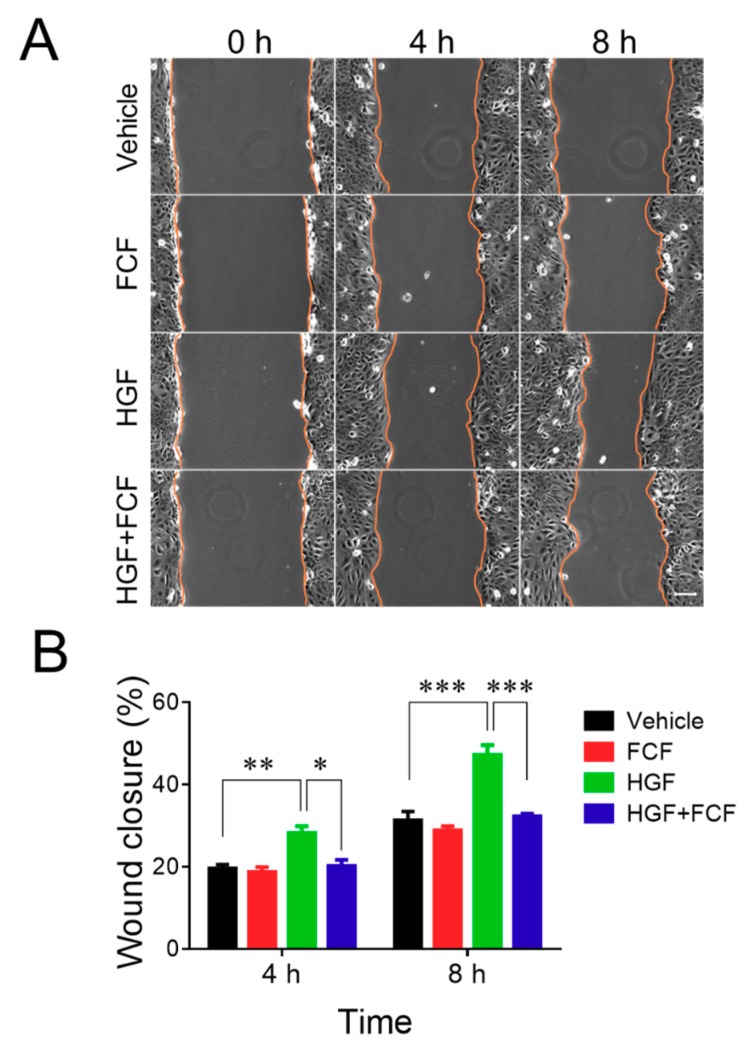
Forchlorfenuron attenuates hepatocyte growth factor-induced migration of prostate epithelial cells. Confluent DU145 cell monolayers were pretreated for 2 h with either FCF (50 μM), or vehicle (DMSO), and wounded. Spontaneous and HGF (25 ng/mL)-induced wound closure with, or without, FCF was examined at the indicated time points. (**A**) Representative images of wounded DU145 cell monolayers. (**B**) Quantitation of wound closure during 4 and 8 h of cell migration. Data are presented as a mean ± SE (*n* = 5); **p* < 0.05, ***p* < 0.01, ****p* < 0.001. Scale bar, 100 µm.

**Figure 3 cells-09-00084-f003:**
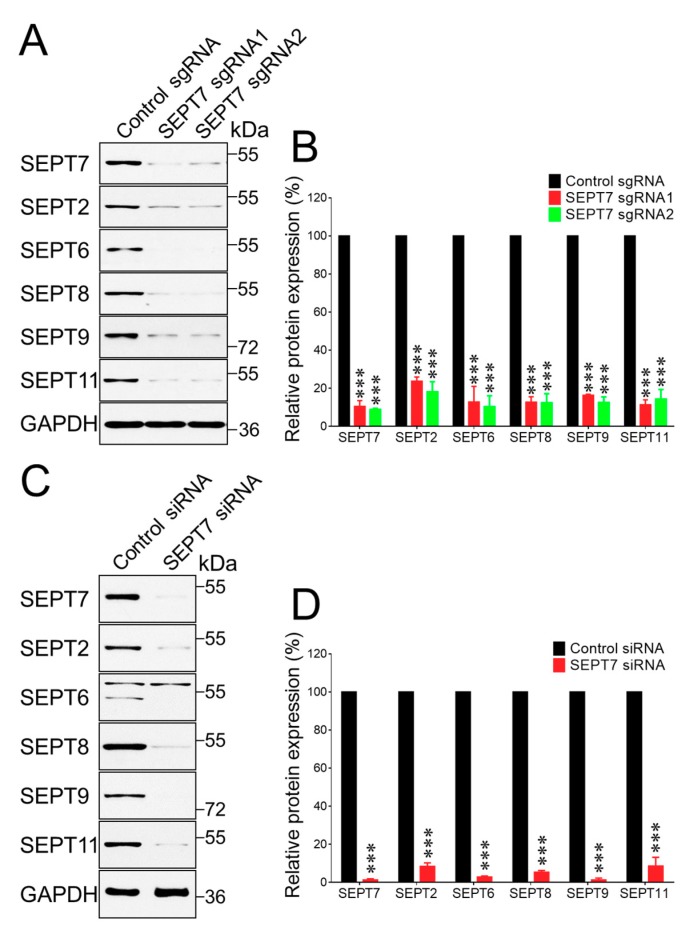
Either CRISPR/Cas9-mediated knockout or siRNA-mediated knockdown of SEPT7, markedly decreases the expression of other septin proteins in epithelial cells. SEPT7 was either knocked out in HT-29 cells using CRISPR/Cas9 mediated gene editing using two different single guide RNAs (sgRNA1 and sgRNA2, **A**,**B**) or knocked down in DU145 cells using a siRNA SmartPool (**C,D**). Representative immunoblots (**A**,**C**) and densitometric quantification of septin expression (**B**,**D**) are shown. Data are presented as a mean ± SE (*n* = 3); ****p* < 0.001.

**Figure 4 cells-09-00084-f004:**
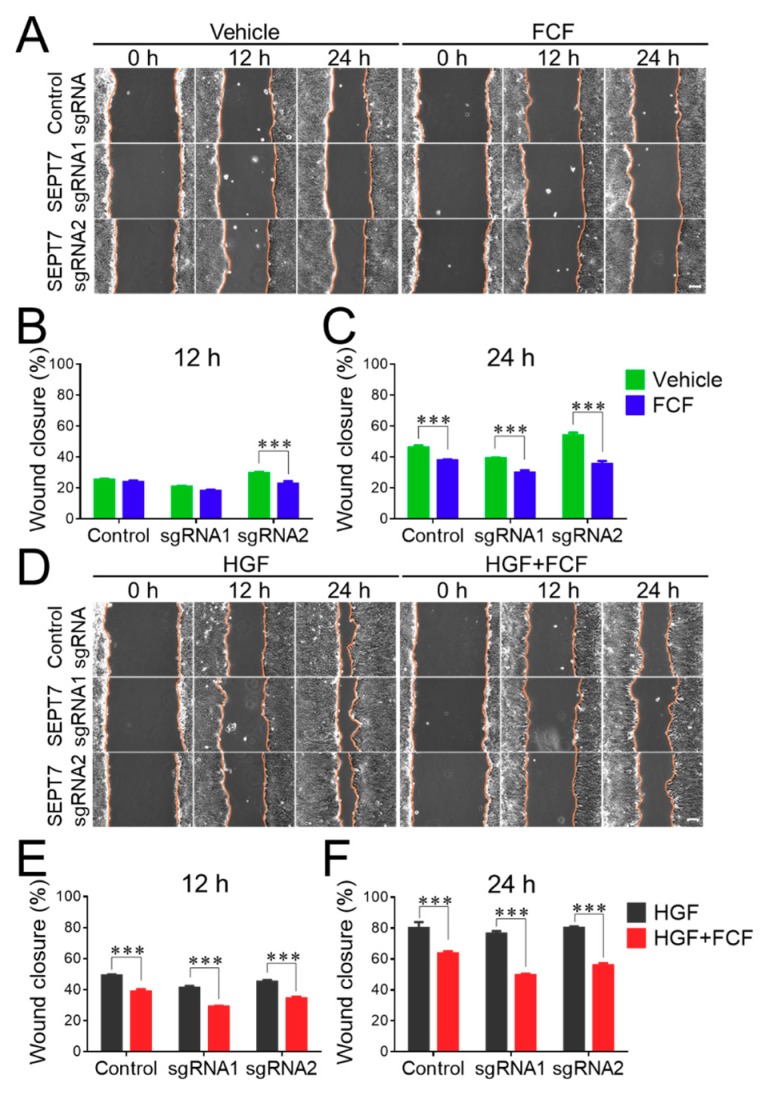
Forchlorfenuron inhibits spontaneous and hepatocyte growth factor-induced migration of control and septin-deficient colonic epithelial cells. HT-29 cells with stable CRISPR/Cas9 mediated knockout of SEPT7 and control cell monolayers were subjected to either spontaneous or HGF-induced wound healing assay with, and without, FCF, as described in the Figure 1 legend. (**A**,**D**) Representative images of wounded HT-29 cell monolayers. (**B**,**C**,**E**,**F**) Quantitation of wound closure during 12 and 24 h of cell migration. Data are presented as a mean ± SE (n = 5); ****p* < 0.001. Scale bars, 100 µm.

**Figure 5 cells-09-00084-f005:**
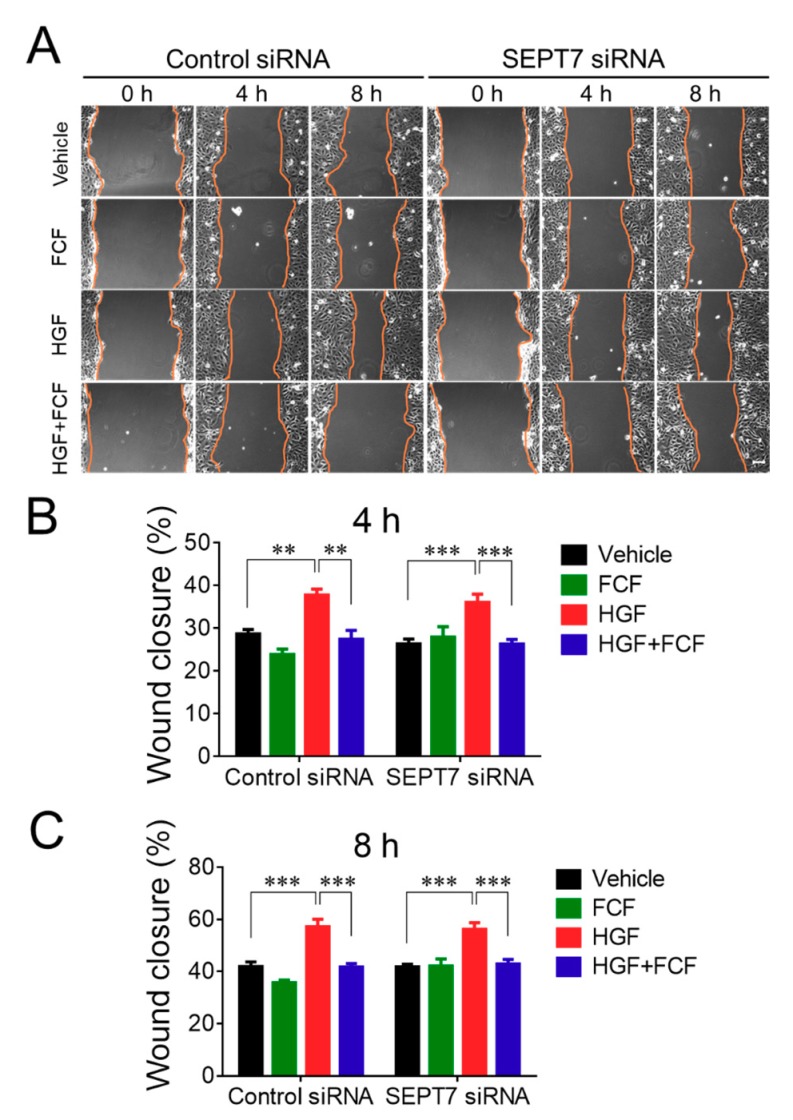
Forchlorfenuron inhibits hepatocyte growth factor-induced migration of control and septin-depleted prostate epithelial cells. Control and SEPT7-depleted DU145 cells were subjected to either spontaneous or HGF-induced wound healing assay with, and without, FCF, as described in the Figure 2 legend. (**A**) Representative images of wounded DU145 cell monolayers. (**B**,**C**) Quantitation of wound closure during 4 and 8 h of cell migration. Data are presented as a mean ± SE (n = 5); ***p* < 0.01, ****p* < 0.001. Scale bar, 100 µm.

**Figure 6 cells-09-00084-f006:**
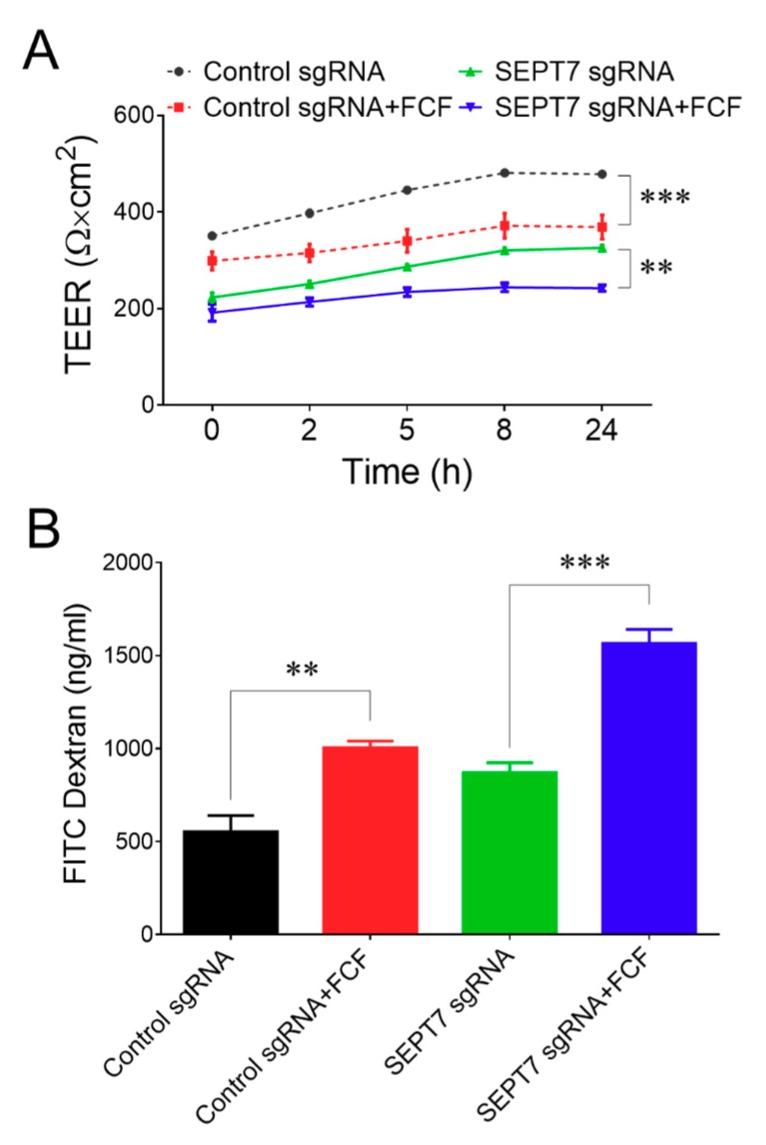
Forchlorfenuron disrupts barrier integrity in colonic epithelial cell monolayers. Control and SEPT7-depleted HT-29 cell monolayers were treated with either FCF (50 μM) or vehicle. Epithelial barrier integrity was examined by measuring transepithelial electrical resistance (TEER) at the indicated times (**A**) or by measuring transepithelial FITC-dextran flux after 24 h of FCF treatment (**B**). Data are presented as mean ± SE (*n* = 3); ***p* < 0.01, ****p* < 0.001.

**Figure 7 cells-09-00084-f007:**
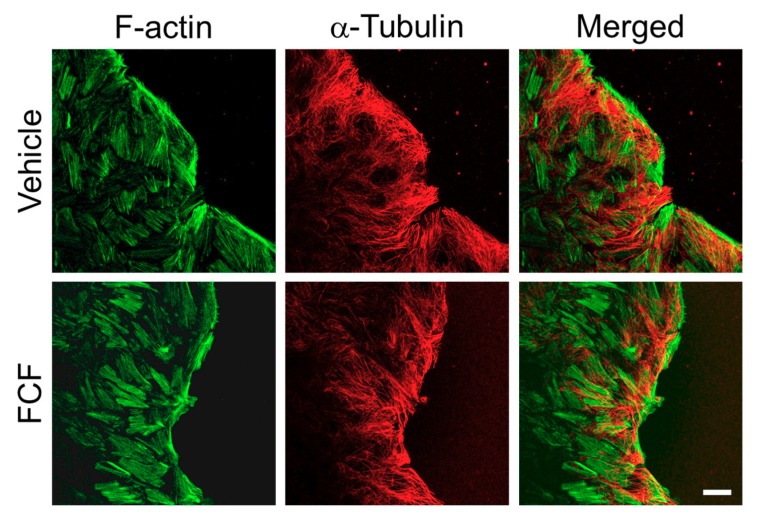
Forchlorfenuron does not affect the actin or microtubule cytoskeleton in migrating colonic epithelial cells. Control and SEPT7-depleted HT-29 cell monolayers were wounded and allowed to migrate for 12 h. Cells were fixed and subjected to dual fluorescence labeling for F-actin (green) and α-tubulin (red). Scale bar, 20 µm.

**Figure 8 cells-09-00084-f008:**
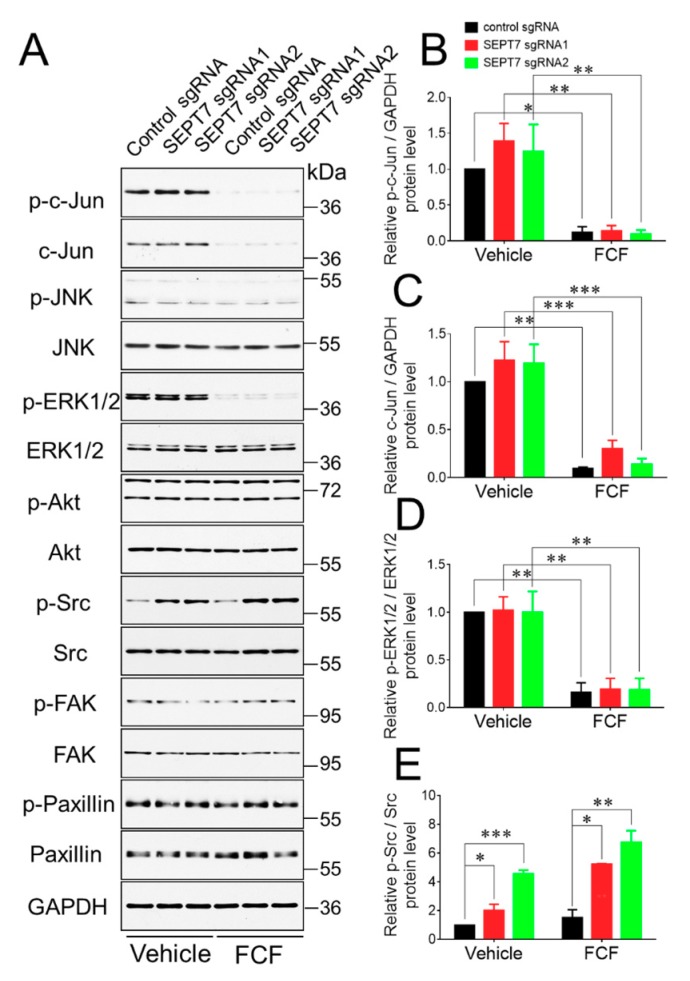
Forchlorfenuron treatment affects different signaling events in migrating colonic epithelial cells. Control and SEPT7-depleted HT-29 cell monolayers were subjected to multiple wounding and allowed to migrate for 12 h in the presence of either FCF (50 μM) or vehicle. Total cell lysates were used for immunoblotting analysis to evaluate the activation of different signaling pathways. Representative immunoblots (**A**) and densitomentic quantification of protein expression (**B**–**E**) are shown. Data are presented as a mean ± SE (*n* = 3); **p* < 0.05, ***p* < 0.01, ****p* < 0.001.

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
