# Peer review of "A Septin Cytoskeleton-Targeting Small Molecule, Forchlorfenuron, Inhibits Epithelial Migration via Septin-Independent Perturbation of Cellular Signaling"

_cells, 2019, doi:10.3390/cells9010084_

Round 1
Reviewer 1 Report
This manuscript entitled “A septin cytoskeleton-targeting small molecule, forchlorfenuron, inhibits epithelial migration via septin-independent perturbation of cellular signaling” examined the specificity of the forchlorfenuron (FCF), a plant cytokinin, on septin cytoskeleton and possible off-target effects in HT-29 and DU145 human epithelial cells. The authors demonstrate and conclude that the modulator of the septin cytoskeleton, FCF, potently inhibits migration and impairs the barrier properties of human epithelial cells in a septin cytoskeleton-independent fashion. Thus, they claim that his is the first evidence of non-septin (i.e. off-target) effects of FCF in mammalian cells.
This is a well written, interesting, and present a useful contribution. The methods are really detailed and appropriate for this study, the results are satisfying, and the discussion is extensive and well-written. This study was well prepared, and I would like to congratulate the authors for their effort and time. However, I have a few comments that are as follows:
Please, in the experiments of cell migration (Result 3.1), to explain why the differences of treatment times between the cells (HT-29, times of 12 and 24 hs; and DU145, times of 4 and 8 hs). Also, in this same experiment, will be interesting to discuss why the FCF treatment affected spontaneous cell migration in HT-29 cell monolayers but not in DU145 cell monolayers. Downregulation of septin 7 expression (Result 3.2). Why did not use CRISP/Cas9 to SEPT7 downregulation in both cells?? I think that this should be justified in the text. In Results 3.3, the authors decided also to investigate the effects of the FCF compound on epithelial barrier permeability. Considering that this compound is involved in others off-target effects, as specify in the discussion section, I consider that the decision to analyze the barrier permeability must be justified. In the experiments where don’t not use DU145 cells, e.g. paracellular permeability, immunofluorescence to actin filaments and microtubules and cell signaling proteins, please to justify why you didn't use these cells.
I hope that my comments are useful for the improvement of the article.
Author Response
Reviewer 1.
We thank the reviewer for a very positive evaluation of our study and valuable comments that are addressed below:
Comment 1: Please explain why the different treatment time was used for HT-29 and DU145 cells.
Response: The time difference between these two cell lines reflects the different migration rates of these cells. For example, DU145 cells migrate fast, and a 4-8 h time frame was sufficient to achieve significant wound healing in control and FCF-treated DU145 cell monolayers. In contrast, HT-29 are slowly-migrating cells, thus a prolonged time point (12-24 h) was required to achieve 20-50% of wound closure in HT-29 cell monolayers. We added this explanation to the Result section 3.1 of the revised manuscript.
Comment 2: Why FCF treatment affected spontaneous migration of HT-29, but not DU145 cells?
Response: While we do not know the exact reasons for the differential sensitivity of HT-29 and DU145 cells to FCF, it may be due to differences in the dose-responsiveness of these cells to the drug. We selected one of the lowest effective concentrations of FCF (50 micromoles) for the wound healing experiments. This concentration was sufficient to attenuate HGF-induced motility of DU145 cells, however higher concentrations of FCF inhibition could be required to block spontaneous migration of these cells.
Comment 3: Why CRISPR/Cas9-mediated knockout of SEPT7 was not used for both HT-29 and DU145 cells?
Response: We disrupted the septin cytoskeleton with stable CRISPR/Cas9-dependent knockout of SEPT7 in HT-29 cells and transient siRNA-mediated knockdown of SEPT7 in DU145 cell. There are two reasons for using different approaches to deplete SEPT7. First, CRISPR/Cas9 gene editing and RNA interference may have their own, specific off-target effects in targeted cells. The fact that both approaches for SEPT7 depletion yield similar responses to FCF treatment suggests that these responses are a direct consequence of septin depletion. Second, stable CRISPR/Cas9-mediated gene editing could be accompanied by cell compensatory responses, which do not occur during transient siRNA-mediated gene knockdown. We have now added this explanation to page 6 of the revised manuscript.
Comment 4: Examining epithelial barrier should be better justified. Why only HT-29 cells were used to study the epithelial barrier?
Response: Since both cell migration and establishment of the epithelial barrier are mediated by similar molecular mechanisms, such as cross talk between the septin cytoskeleton and other cytoskeletal structures, it was logical to add barrier examination as another functional readout in this study. Furthermore, FCF was previously shown to disrupt the epithelial barrier (Sidhaye VK et al Am J Resp Cell Mol Biol 2011, 45 120-26). The focus on HT-29 cells in this study was because, in our hands, DU145 cells did not develop strong paracellular barrier. We now provide more explanation on page 8 of the revised manuscript.
Comment 5: Why DU145 were not used to examine the effects of FCF on the cytoskeleton and cellular signaling?
Response: Since FCF did not inhibit the spontaneous migration of DU145 cells we did not include these cells in the immunocytochemical and biochemical experiments aimed at examining the mechanisms underlying FCF-dependent inhibition of spontaneous epithelial cell migration.
Reviewer 2 Report
The paper is logical concise and well written, presenting an important new evidence on an unexpected mechanism of action for what is thought to be a septin-targeting small molecule, FCF. Data are well presented and described and support the main conclusion.
This reviewer has several suggestions that could improve the impact of this work.
Authors conclude that FCF inhibition of c-jun leads to diminished migration. Authors previously showed that disassembly of the apical IEC junctions also involves c-jun activation. As such, could the observed effects of FCF on IEC permeability be also mediated by c-jun? Inhibition experiments would prove this point and if rescue migration would also prove causality for both processes. It is also of interest which, if at all of the junctional proteins regulating barrier are effected by FCF treatment.Minor, legend to figure 4 missing panels E/F
line 194: recommend adding HGF-stimualted migration.... and removing the commas. as is sentence is confusing.
Figure 7 would benefit from better resolution images of actin. perhaps zoom-in could be added.
Author Response
Reviewer 2.
We thank the reviewer for finding this paper logical concise and well-written and for providing insightful comments that we address below:
Comment 1: Authors previously shown that disassembly of apical IEC junctions also involves c-jun activation. Could the observed effects of FCF on IEC permeability be also mediated by c-Jun?
Response: We are pleased that the reviewer is familiar with our previous study demonstrating that activation of c-Jun-dependent N-terminal kinase (JNK) disrupted model intestinal epithelial barrier. While downregulation of c-Jun is different from JNK activation, it may underline the observed disruption of HT-29 cell barrier during FCF exposure. We now mention this possibility on page 14 of the revised manuscript.
Comment 2: It would also be of interest which junctional proteins regulating barrier are affected by FCF treatment.
Response: We appreciate this insightful question. Given our exciting findings that genetic disruption of the septin cytoskeleton in SEPT7-knockout cells increases the permeability of HT-29 cell barrier, we believe that it would be interesting to elucidate the underlying mechanisms, including the effects of septin depletion on the expression of different junctional proteins. We have a separate ongoing study that targets this important question and the results of this study will be published elsewhere.
Comment 3: Legend of Figure 4 does not mentioned E/F panels.
Response: we apologize for this error, which has been corrected.
Comment 4: Some text editing is recommended on line 194.
Response: We appreciate and incorporate these suggestions.
Comment 5: Figure 7 would benefit from better resolution image of actin. Perhaps zoom in could be added.
Response: HT-29 are small cells with short actin filament bundles. It is not easy to image these actin cytoskeletal structures using conventional confocal microscopy. The presented images have been acquired at highest possible magnification. We tried to include zoomed images, but those appeared relative fuzzy and informative. We opted therefore to replace this Figure and include higher intensity F-actin images that better represent the overall of organization of the actin cytoskeleton in control and FCF-treated cells.
Reviewer 3 Report
This submission demonstrated that a synthetic small molecule forchlorfenuron (FCF) inhibits spontaneous, as well as hepatocyte growth factor-induced, migration in HT-29 and DU145 human epithelial cells. I like to give the following comments.
FCF is widely used to probe septin functions. Is it the same concentration for using to interfere with cell proliferation and others? Pathophysiologic role of the septin-independent functions of FCF may speculate in the introduction section. Lentiviruses produced in HEK293T cells that needs to show the source. Additionally, the Feng Zhang Lab needs the background(s). In Figure 5, wound healing assay estimated within 8 hours in DU145 cell that varied with assay in HT-29 cell. Why? In Figure 7, quantification of the difference between FCF-treated group and Vehicle-treated control may support the title. As a pharmacological modulator, dose-dependent effect of FCF seems important. Not use of FCF alone is one of the key-points in this report. Concern(s) of the used dose may support it. Limitation(s) of the results may strengthen the discussion.Author Response
Reviewer 3.
We appreciate the valuable comment by this reviewer, which are addressed below.
Comment 1: Is the concentration of FCF used in this study the same as was used by others?
Response: In our study, we used 50 micromoles of FCF, which is at the lower end of the concentration range (50-500 micromoles) used in previously published studies. We now state this on page 5 of the revised manuscript.
Comment 2: Pathophysiologic roles of the septin-independent functions of FCF may be speculated in the introduction.
Response: Based on previous publications, we described several cellular/tissue effects of FCF in the Introduction and Discussion parts of the paper. Since septin-dependent and independent effects of FCF have not been previously well-defined or delineated, it is difficult to speculate about the physiological or pathophysiological implications of septin-independent actions of this compound. This important question requires further clarification.
Comment 3: Source of lentiviral-producing HEK293T cells should be indicated.
Response: We apologize for this omission and now describe the source of lentivirus-packaging 293FT cells.
Comment 4: The Feng Zhang lab needs the background.
Response: affiliation of Dr. Zhang’s lab is now included.
Comment 5: Why different time points were used to examine HT-29 and DU145 cell wound healing.
Response: Different time points were selected due to different wound healing rates in these two cell lines. Please see our response to the Comment 1 from the Reviewer 1.
Comment 6: Quantification is requested for the difference between vehicle and FCF-treated groups in Figure 7.
Response: Figure 7 shows the lack of major effects of FCF on the organization of actin filaments and microtubules at the migrating cell edge. It is difficult to provide a general quantification of these images, since the derived information involves not only the signal intensities of the labeled proteins but also their localization patterns. Since the presented data shows no effects of FCF on the described cytoskeletal structures, we do not believe that quantification of the presented representative images could significantly strengthen this conclusion. We hope the reviewer will accept our arguments.
Comment 7: The reviewer highlights the importance of dose-effect of FCF treatment.
Response: We completely agree with this reviewer’s assessment. However, FCF is known to be a low affinity ligand for septins that is active in the tens-hundreds micromolar range. In fact, the majority of published studies used 100 micromoles of FCF. Our study shows that even a lower FCF concentration of 50 micromoles is sufficient to cause major off-target effects in epithelial cells. Therefore, it is expected that higher FCF concentrations will have even more off-targets effects in different experimental systems.